# Measuring the frequency chirp of extreme-ultraviolet free-electron laser pulses by transient absorption spectroscopy

Thomas Ding [1✉], Marc Rebholz[1], Lennart Aufleger[1], Maximilian Hartmann [1], Veit Stooß[1], Alexander Magunia[1], Paul Birk[1], Gergana Dimitrova Borisova[1], David Wachs[1], Carina da Costa Castanheira[1], Patrick Rupprecht [1], Yonghao Mi[1], Andrew R. Attar[2], Thomas Gaumnitz [3], Zhi-Heng Loh[4], Sebastian Roling[5], Marco Butz[5], Helmut Zacharias[5], Stefan Düsterer [6], Rolf Treusch [6], Arvid Eislage[6], Stefano M. Cavaletto [1], Christian Ott [1✉] & Thomas Pfeifer [1✉]

High-intensity ultrashort pulses at extreme ultraviolet (XUV) and x-ray photon energies, delivered by state-of-the-art free-electron lasers (FELs), are revolutionizing the field of ultrafast spectroscopy. For crossing the next frontiers of research, precise, reliable and practical photonic tools for the spectro-temporal characterization of the pulses are becoming steadily more important. Here, we experimentally demonstrate a technique for the direct measurement of the frequency chirp of extreme-ultraviolet free-electron laser pulses based on fundamental nonlinear optics. It is implemented in XUV-only pump-probe transient-absorption geometry and provides in-situ information on the time-energy structure of FEL pulses. Using a rate-equation model for the time-dependent absorbance changes of an ionized neon target, we show how the frequency chirp can be directly extracted and quantified from measured data. Since the method does not rely on an additional external field, we expect a widespread implementation at FELs benefiting multiple science fields by in-situ on-target measurement and optimization of FEL-pulse properties.

[1] Max-Planck-Institut für Kernphysik, Saupfercheckweg 1, 69117 Heidelberg, Germany. [2] Department of Chemistry, University of California, Berkeley, CA 94720, USA. [3] Laboratorium für Physikalische Chemie, ETH Zürich, Vladimir-Prelog-Weg 2, 8093 Zürich, Switzerland. [4] Division of Chemistry and Biological Chemistry, and Division of Physics and Applied Physics, School of Physical and Mathematical Sciences, Nanyang Technological University, Singapore 637371, Singapore. [5] Physikalisches Institut, Westfälische Wilhelms-Universität Münster, Busso-Peus-Straße 10, 48149 Münster, Germany. [6] Deutsches Elektronen-Synchrotron DESY, Notkestraße 85, 22607 Hamburg, Germany. ✉email: thomas.ding@mpi-hd.mpg.de; christian.ott@mpi-hd.mpg.de; thomas.pfeifer@mpi-hd.mpg.de

The rapidly developing free-electron laser (FEL) technology[1–4], delivering brilliant femtosecond and even attosecond pulses at high photon energy[5,6], enables the use of x-rays for state-selective and phase-sensitive multidimensional spectroscopy[7] and coherent control[8]. The direct and routine measurement of the spectral phase of currently available extreme ultraviolet (XUV) and x-ray FEL pulses is the key to fully implement such nonlinear coherent control concepts in order to find and set optimized pulse parameters for their interaction with matter. Direct temporal diagnostic tools[9] for self-amplified spontaneous emission (SASE) XUV/x-ray FEL pulses are the methods of linear[10] and angular[11] streaking, which are sensitive to the temporal shape of the pulses, also including the chirp. These methods rely on the availability of a temporally synchronized and sufficiently intense external field. A complementary route to diagnosing the temporal structure of SASE-radiation pulses is by measuring the FEL-lasing-induced energy losses in the electron bunch (for example with an X-band radio-frequency transverse deflecting cavity (XTCAV)[12]), from which the temporal profile of the XUV/x-ray emission can be reconstructed. For the case of seeded FEL pulses, the generation of two almost identical FEL pulses and the evaluation of their XUV interferogram allows for a full characterization of their spectro-temporal content[13].

In this work, we present a technique to directly measure the XUV-FEL frequency chirp without relying on any additional external field or seeded multi-pulse schemes. Since the reported technique provides on-target access to the spectro-temporal distribution of the XUV radiation, it is ideal for in situ diagnosing of user experiments with sensitivity to the FEL lasing performance. For instance, here, we experimentally observe a systematic dependence of the frequency chirp on the FEL pulse energy (increasing chirp for decreasing pulse energy).

## Results

**Frequency-resolved plasma gating.** The basic concept of this metrology scheme is adopted from femtosecond transient absorption spectroscopy with chirped super-continuum probing[14]. In those measurements, the white-light continuum (WLC) is typically strongly chirped due to its nonlinear generation process and dispersion of optical elements and can be well-characterized and compensated for in experimental measurements. Commonly, one measures the nonlinear absorption signal as a function of the time delay between the pump and the WLC-probe broadband laser pulses while spectrally resolving the WLC-probe photon energy. For instance, the pump-induced nonlinear interaction with the sample produces a rapid change of its refractive index and thus switches[15–17] the WLC transmission of the probe pulse. Correlating the measured WLC probe spectra with the time delay then allows for tracking the arrival time of the different WLC frequency components at the sample. This concept has previously been realized for the (x-ray-) FEL-induced change of the refractive index in the visible to near-infrared part of the electromagnetic spectrum, which is probed by an optical laser pulse as a function of time delay. This allows both for a few-femtosecond[18] and sub-femtosecond[19] measurement of the relative timing between laser and FEL pulses, commonly known as the timing tool, as well as for monitoring the FEL-pulse duration[20].

Here, we propose ultrafast plasma generation as an ultrafast gate in combination with spectrally resolved absorption measurements to precisely quantify chirped pulses in the XUV. As a specific example for demonstration, we utilize a well-characterized fundamental nonlinear process of atomic physics for XUV-optical gating: the sequential double ionization of neon via two XUV photons, which has been extensively investigated[21–23]. The measurement (see Fig. 1a) is carried out through XUV-pump and

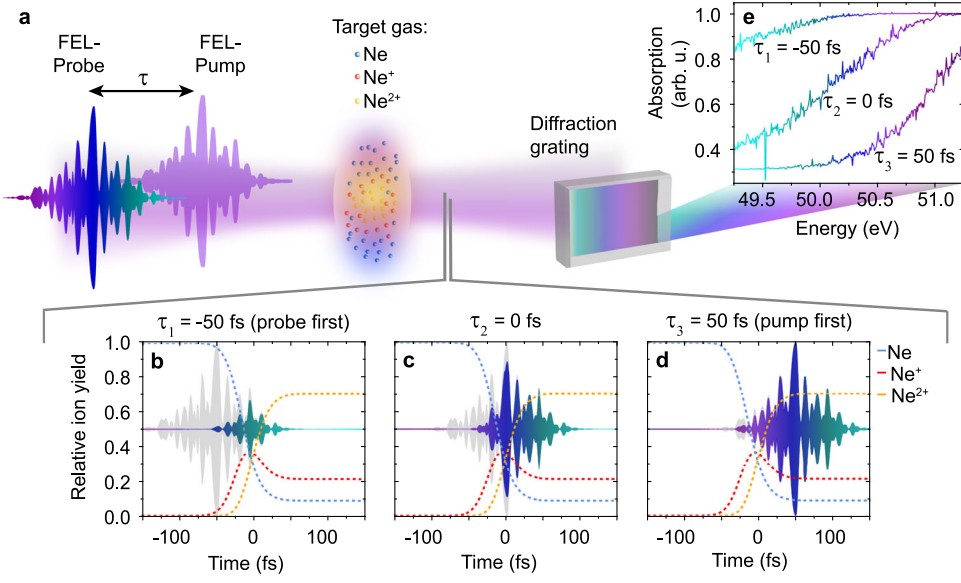

**Fig. 1 Principle of frequency-resolved plasma gating. a** Measurement scheme using XUV double ionization of a neon gas target [depicted by blue, red, and yellow spheres, respectively indicating neutral neon (Ne), and its singly charged (Ne⁺) and doubly charged (Ne²⁺) ions] for frequency-resolved gating of a chirped XUV-FEL probe pulse. This leads to a time-delay-dependent change of the probe-pulse absorption spectrum measured with a diffraction grating. **b–d** Averaged time-dependent relative atomic/ionic population of Ne, Ne⁺, and Ne²⁺ calculated using 50-fs SASE pump pulses[24] with, on average, $4 \times 10^{13}$ Wcm⁻² peak intensity. The transmitted (gated) temporal pulse profile of a chirped SASE probe pulse and the respectively passed frequency band are illustrated by violet, blue, and green-colored SASE "spikes" (different frequency components are represented by different colors) for three different time-delay settings: $\tau_1 = -50$ fs (probe first), $\tau_2 = 0$ fs (pump-probe pulse overlap), $\tau_3 = 50$ fs (pump first). In gray color, the gated probe-pulse profiles are depicted, attenuated through absorption in the initially abundant Ne and Ne⁺ target species. **e** Corresponding transient absorption spectra showing a delay-dependent modification of the absorption (or gate) band for varied time delays ($\tau_1$, $\tau_2$, and $\tau_3$).

XUV-probe transient absorption in transmission geometry, where the XUV pulses are centered at a photon energy of about 50 eV. More specifically, the XUV-FEL pump pulse, which represents the gate pulse, efficiently doubly ionizes the moderately dense neon gas target and thus promptly turns the medium from a highly opaque into a highly transparent state, which modifies the spectrally resolved transmission of the subsequent XUV-FEL probe pulse. The latter represents the XUV analog of a broadband optical continuum with a typical SASE spectral bandwidth of 0.5–2% at full width at half maximum (FWHM) with respect to the central photon energy. This switching mechanism from opaque to transparent is the result of an abundant (on the order of several 10%) aggregation of $Ne^{2+}$ ions within the pumped target volume. While Ne and $Ne^+$ are strongly absorbing (cross sections about 9 Mbarn at 50 eV), $Ne^{2+}$ and all higher charge states are highly transparent since the absorption of a single 50-eV photon is not sufficient for direct ionization. This ionization-induced transmission gate thus effectively acts as an ultrafast shutter on the XUV-FEL probe pulse switching states between open (low abundance of Ne and $Ne^+$) and closed (high abundance of Ne and $Ne^+$). The time at which the shutter opens for the probe pulse can be controlled by the time delay $\tau$ with respect to the ionizing pump pulse; the shutter speed is hence given by the Ne and $Ne^+$ depletion time, which is here attributed to the sequential double-ionization dynamics triggered by the pump pulse. If the target depletion time is sufficiently short (i.e., on the order of the probe-pulse duration), the step-like transmission gate function serves as a "temporal knife-edge[16,17]" slicing the XUV-FEL probe-pulse as soon as the shutter gate opens. For frequency-chirped probe pulses, the temporal gate is equivalent to a frequency gate, passing all the spectral components of the radiation field during the open period of the shutter, and suppressing those spectral components that arrive during the close period. This time-domain gating principle is illustrated in Fig. 1b–d for the case of ~70% final $Ne^{2+}$ population, and the corresponding spectrally resolved transmission changes are shown in Fig. 1e. Thus, the technique proposed here to directly measure the frequency chirp of XUV-FEL pulses consists of measuring the frequency-($\omega$)-resolved transient absorption signal of the probe pulse and tracing the step-like increase in transmission as a function of time delay $\tau$ with respect to the pump pulse. Hereby the statistical ensemble of SASE FEL pulses can be modeled with the partial-coherence method[24], which assumes a fluctuating spectral phase, on top of which an average frequency chirp can be added (see "Computational model" and Fig. 5 for details). Only considering this average chirp, the time delay versus frequency relationship is mathematically expressed by

$$\tau(\omega) = \Phi'(\omega) \approx D_1 + D_2(\omega - \omega_L) + \frac{1}{2}D_3(\omega - \omega_L)^2 + \frac{1}{6}D_4(\omega - \omega_L)^3 + \ldots \quad (1)$$

which can be directly integrated to obtain the spectral phase

$$\Phi(\omega) \approx D_1(\omega - \omega_L) + \frac{1}{2}D_2(\omega - \omega_L)^2 + \frac{1}{6}D_3(\omega - \omega_L)^3 + \frac{1}{24}D_4(\omega - \omega_L)^4 + \ldots \quad (2)$$

and used to fit the dispersion coefficients $D_n$ for orders $n \geq 1$. Notice that the dispersion coefficients are determined by evaluating the $n$th derivative of $\Phi(\omega)$ with respect to $\omega$ at the center frequency $\omega_L$. The first-order dispersion coefficient, $D_1$, describes a global temporal delay offset, $D_2$ a linear frequency sweep, and all higher orders of dispersion ($D_n$ for orders $n \geq 3$) lead to more complex modifications of the average pulse properties.

**Conceptual demonstration, model simulations**. Using a computational model based on rate equations (see "Computational model"), we now demonstrate how the average frequency chirp of partially coherent XUV-FEL pulses can be extracted from transient absorption spectra. We calculate the absorbance $A(\tau, \omega)$

employing 50-eV SASE pulses with an on-average temporal duration of 50 fs and FWHM spectral bandwidth of the averaged spectrum of ~1 eV. The modeled stochastic pulses vary from shot to shot in their spiky substructure due to a random relative phase for each of them, while a continuous spectral-phase offset according to Eq. (2) is equally inherent to all pulses (cf. "Methods," Fig. 5). We simulate four hypothetic cases of different linear and nonlinear frequency chirps, where the results are shown in Fig. 2a–d. The frequency modulation is computed in an ad hoc manner from both linear and nonlinear dispersion coefficients, which are fed into the power series of the spectral phase expressed by Eq. (2) and serve as input data for the modeling of the pulses (see "Methods"). For the retrieval of the dispersion coefficients from calculated or measured absorbance data, $A(\tau, \omega)$, an error-function fit is applied along the time-delay ($\tau$) axis at each $\omega$, in order to pinpoint the spectro-temporal position of the transient increase in transmission (related to the position at half of the amplitude). This directly yields the dispersion function $\tau(\omega)$ as expressed by Eq. (1) and thus contains all orders $n$ of the dispersion coefficients $D_n$. The latter can be extracted by fitting $\tau(\omega)$ with a polynomial model function. Figure 2e–h shows the input dispersion functions $\tau(\omega)$ of the simulation as red solid curves; the blue dots are the associated fitted (to error function in $\tau$) data points from the calculation results of Fig. 2a–d, showing excellent agreement. For all example cases discussed here, the deviation between the reproduced dispersion coefficients and the input data is determined at around 1% (see Table 1). Inaccuracies can be further minimized by increasing the time- and photon-energy resolution as well as the statistical sampling.

**Experimental realization**. Next, we demonstrate the direct applicability of the technique in an experiment carried out at the free-electron laser in Hamburg (FLASH). Commensurate to the computational model, the FEL is operated at ~50 eV photon energy (~1 eV averaged FWHM spectral width), 50–100-fs pulse duration and ~50-µJ pulse energy corresponding to on-target intensities in the mid $10^{13}$-Wcm$^{-2}$ range in the ~25-µm (FWHM) diameter focal spot. The accelerator was operated at a moderate compression, chirping the electron bunch (for details, see "Supplementary Note 1, Accelerator settings and performance") in order to provide a broad XUV photon bandwidth for transient-absorption measurements.

The beamline setup includes an XUV autocorrelator[25] and allows for the preparation of XUV-pump and time-delayed XUV-probe FEL pulses, which are structured almost identically in time (see "Experimental beamline setup"). While the pump and probe pulses spatially overlap within the focal interaction volume, they are again separated in the far field behind the focus and the transmitted spectrum of the probe pulse, denoted by $I_{pr}(\omega)$, can be measured separately. Spectra taken before passing through the neon target are termed reference spectra, $I_{ref}(\omega)$, and were measured simultaneously using the online photon spectrometer at FLASH[26]. The absorbance (optical density (OD)) of the probe pulse is then determined via

$$A_{exp}(\tau, \omega) = -\log_{10}\left\{\frac{\langle I_{pr}(\tau, \omega)\rangle}{\langle I_{ref}(\omega)\rangle}\right\}, \quad (3)$$

where the mean value $\langle \ldots \rangle$ over about 500 individual (single-shot) spectra is taken for each time-delay setting. In Fig. 3, we compare the measurement of the absorbance $A_{exp}(\tau, \omega)$ with numerical results. A spectrally dependent shift of the absorbance drop is clearly visible, which agrees well with the above discussion of the transmission of a chirped pulse through an ionizing time-dependent gate. The retrieved dispersion function $\tau_{exp}(\omega)$ through error-function curve fitting (see

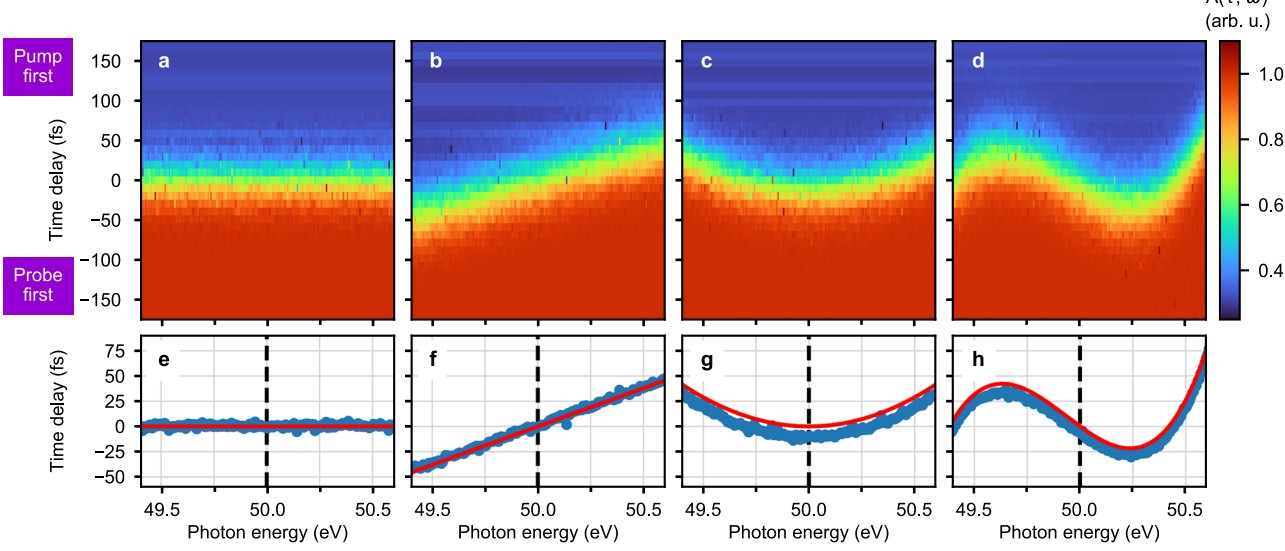

**Fig. 2 Numerical simulation of the effect of plasma gating. a–d** Calculated neon absorbance spectra for 50-fs SASE pulses, having averaged over 200 single SASE-FEL pulses per time-delay setting. Simulation for different values of $D_n$ for $n \geq 2$ with increasing order from left to right. **e–h** The red solid curve represents the dispersion function [$\tau(\omega)$, see Eq. (1)], which was used as input for the calculation of the absorbance and corresponding output data (blue dots) are extracted by fitting a step function to the calculation results. The dashed vertical line indicates the 50-eV center photon energy (bandwidth around 1 eV). In the trivial case of no dispersion, i.e., $D_n = 0$ for $n \geq 2$, displayed in **a**, **e**, the absorption spectrum is governed by a time-dependent "ramp" signal located at around $\tau = 0$ (pump-probe pulse overlap) and is constant over $\omega$. The case of a second-order (group-delay) dispersion of $D_2 = 50 \text{ fs}^2$, displayed in **b**, **f**, yields a steep linear dependence of $\tau(\omega)$. **c**, **g** Parabola-shaped signal trace obtained through imposing a third-order dispersion of $D_3 = 100 \text{ fs}^3$, and, displayed in **d**, **h**, the "wavy"-shaped signal trace obtained through a dominant fourth-order dispersion (combined dispersion coefficients $D_2 = -100 \text{ fs}^2$, $D_3 = 100 \text{ fs}^3$ and $D_4 = 1000 \text{ fs}^4$). See Table 1 for quantitative analysis of numerical input and retrieved data.

**Table 1 Numerical input and retrieved data for different configurations of spectral phase to quantify the retrieval accuracy.**

|  | Input data | Retrieved data |
|---|---|---|
| Case a, e |  |  |
| $D_2$ | $0.0 \text{ fs}^2$ | $0.2 \pm 0.7 \text{ fs}^2$ |
| $D_3$ | $0.0 \text{ fs}^3$ | $0.0 \pm 1.0 \text{ fs}^3$ |
| $D_4$ | $0.0 \text{ fs}^4$ | $-1.5 \pm 4.6 \text{ fs}^4$ |
| Case b, f |  |  |
| $D_2$ | $50.0 \text{ fs}^2$ | $49.3 \pm 0.6 \text{ fs}^2$ |
| $D_3$ | $0.0 \text{ fs}^3$ | $0.5 \pm 0.7 \text{ fs}^3$ |
| $D_4$ | $0.0 \text{ fs}^4$ | $2.6 \pm 2.9 \text{ fs}^4$ |
| Case c, g |  |  |
| $D_2$ | $0.0 \text{ fs}^2$ | $-0.3 \pm 0.4 \text{ fs}^2$ |
| $D_3$ | $100.0 \text{ fs}^3$ | $99.4 \pm 0.9 \text{ fs}^3$ |
| $D_4$ | $0.0 \text{ fs}^4$ | $-0.2 \pm 2.5 \text{ fs}^4$ |
| Case d, h |  |  |
| $D_2$ | $-100.0 \text{ fs}^2$ | $-100.3 \pm 0.9 \text{ fs}^2$ |
| $D_3$ | $100.0 \text{ fs}^3$ | $98.8 \pm 1.1 \text{ fs}^3$ |
| $D_4$ | $1000.0 \text{ fs}^4$ | $1005.9 \pm 6.7 \text{ fs}^4$ |

Tabulated data of cases a–d corresponding to the simulated time-delay traces of Fig. 2a–d and their respective dispersion functions plotted in Fig. 2e–h. Quoted errors are standard deviations of the mean value after ten repetitions of the calculation.

above) is presented in Fig. 3c. We extract a pronounced linear chirp of $D_2 = (32 \pm 1) \text{ fs}^2$, a quadratic chirp of $D_3 = (-8 \pm 1) \text{ fs}^3$ and a cubic chirp of $D_4 = (-40 \pm 5) \text{ fs}^4$. The finding of a predominant linear chirp is consistent with an energy chirp on the electron bunch, which was not fully compressed, while collective space-charge effects are expected to play a minor role (see "Supplementary Note 1, Accelerator settings and performance"). For the simulated data shown in Fig. 3d–f, we used the retrieved dispersion coefficients as input for the numerical model in order to reproduce the experimental data. The good quantitative agreement between

experimental and computational data both demonstrates a high joint spectral (10-meV scale) and temporal (1-fs scale) measurement accuracy and a high fidelity in the retrieval method. We notice that the experiment was performed with a similar pump- vs. probe-pulse intensity sharing of about 2:1. In general, the pump pulse should be the more intense one such that it dominates the ionization process used for the gate. We do not explicitly treat the probe-induced ionization in the numerical modeling, as generally the probe pulse should be set to lower pulse energy than the pump pulse to ensure the latter to dominate the ionization dynamics of the gate. Even a slight ionization offset contribution by the probe pulse would mainly affect the contrast (i.e., amplitude) of the calculated OD-signal ramp, globally for all frequencies, without significantly modifying its shape and thus the extracted spectral-phase coefficients.

**Pulse-energy-resolved analysis.** SASE-FELs produce ensembles of radiation pulses with statistically varying pulse properties, such as the pulse energy, i.e., the number of photons contained in each pulse. In Fig. 4, we make use of this stochastic fluctuation and further quantitatively analyze the frequency chirp as a function of the FEL-pulse energy. We find the trend of increasing spectral chirp (i.e., increasing dispersion coefficients $D_2$, $D_3$, and $D_4$) for decreasing pulse energy. It should be noted, that the retrieved chirp is clearly dominated by the lowest second-order linear contribution related to $D_2$, while the higher orders weigh in less by approximately an order of magnitude (see "Supplementary Note 2, Order-of-magnitude estimation of the nonlinear chirp contributions", for details). Our direct spectral method is thus sensitive to the nonlinear chirp of the measured pulses, evidencing a stronger relative nonlinear contribution [i.e., deviation from a purely linear $\tau(\omega)$] at moderately low fluence (cf. Fig. 4). One possible explanation for this observation may be related to systematics in the process of electron-beam acceleration and compression and its lasing properties, having operated the accelerator not at full compression.

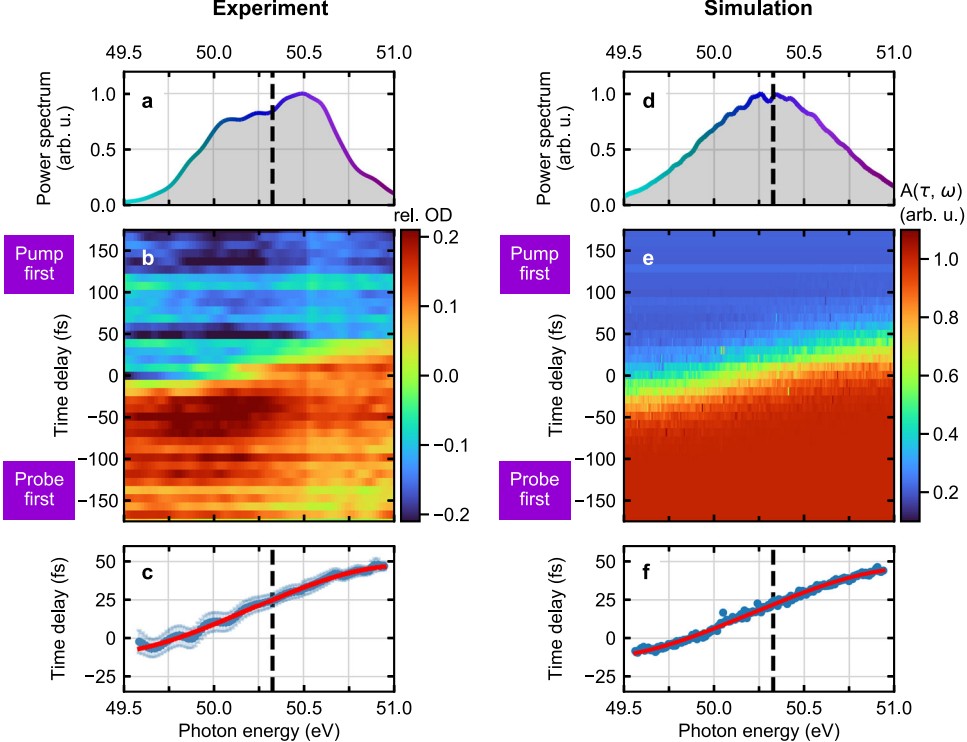

**Fig. 3 Measurement of the XUV-FEL frequency chirp by transient absorption spectroscopy. a** Averaged experimental probe-pulse spectrum without target centered at ~50.3 eV (indicated by the dashed vertical line). **b** Absorbance of the probe spectrum [relative optical density (rel. OD), see "Evaluation of the relative optical density (rel. OD)"] in neon recorded with a spectrometer resolution of ~30 meV and 10-fs time-delay increments. The pump-induced plasma production at positive delays (pump pulse precedes the probe pulse) leads to a prompt drop in the absorbance (increase in transmission) and serves as a "temporal knife-edge". **c** Blue dots, dispersion function $\tau_{exp}(\omega)$ as retrieved through error-function curve fitting the data. Error bars (±5 fs order of magnitude) specify fit errors. Red solid curve, polynomial fit used to determine the individual dispersion coefficients. **d** Averaged simulated probe-pulse spectrum. **e** Simulated absorbance spectrum using the retrieved experimental parameters as input. **f** The red solid line represents the input dispersion function $\tau_{exp}(\omega)$, which perfectly matches the corresponding outcome of the simulation (blue dots).

## Discussion

We have demonstrated an all-XUV-optical technique to characterize the frequency chirp of FEL pulses. Since the technique of transient absorption spectroscopy with XUV SASE-FEL pulses combines a high temporal and spectral resolution on a broad (probe) spectral bandwidth, it can be employed for direct characterization of both low and higher dispersion orders of chirp with high accuracy. Because the method does not require any additional external field, and the same XUV-FEL pulse can be used by means of straightforward spatial beam splitting, it is easy to use and it can be applied in situ during transient absorption experiments. In principle, it is possible to extract the complete phase and amplitude information of the XUV/x-ray FEL pulses from such transient-absorption spectrograms with an optical gate by performing advanced phase-retrieval calculations as recently demonstrated for mid-IR laser pulses[27]. This, however, requires sufficiently well-behaved (phase and amplitude stable) pulses as, e.g., those generated in a seeded FEL[28], while the retrieval is expected to be particularly nontrivial for the case of SASE-FEL pulses, which are composed of temporally coherent sub-pulses with no relative phase correlation between them, and which fluctuate from shot to shot.

Having direct access to the spectro-temporal photon distribution of FEL pulses is a key ingredient for state-selective multi-dimensional spectroscopy at high photon energy[4,29], which now come into reach with next-generation XUV and x-ray FEL radiation sources. The characterization method presented here is not limited to the XUV domain; it can be straightforwardly extended even to the hard x-ray domain, and only requires an x-ray intensity-induced change of the absorbance properties, e.g., through

nonlinear ionization[30], of a moderately dense target medium. In hard-x-ray absorption spectroscopy and when applied specifically to core electrons, a second (intrinsic) shutter is at work due to the Auger effect and the ultrafast refilling of the core hole, which may shut the open gate (transparency gate through core ionization) again quickly on a sub-pulse-duration timescale and may be exploited in the future to further optimize the method.

## Methods

**Computational model.** We aim to calculate the time-dependent abundance of neon in its different ionic charge states and its influence on the absorption. Since our analysis does not involve phase-preserving (coherent) processes, the underlying (density-matrix-based) equation of motion reduces to a set of population-rate equations[31]:

$$\dot{N}_0(t) = -\frac{\sigma_0 N_0(t) I_{pu}(t)}{\hbar \omega_L} \tag{4}$$

$$\dot{N}_1(t) = \frac{\sigma_0 N_0(t) I_{pu}(t)}{\hbar \omega_L} - \frac{\sigma_1 N_1(t) I_{pu}(t)}{\hbar \omega_L} \tag{5}$$

$$\dot{N}_2(t) = \frac{\sigma_1 N_1(t) I_{pu}(t)}{\hbar \omega_L}, \tag{6}$$

where $N_0(t)$, $N_1(t)$, and $N_2(t)$ denote the relative population of neutral neon (Ne) and its singly charged (Ne⁺) and doubly charged (Ne²⁺) ionic species, respectively. The ionization cross sections at photon energy $\hbar\omega_L = 50$ eV for the sequential single-photon absorption steps from Ne → Ne⁺ and Ne⁺ → Ne²⁺ are given by $\sigma_0 = 9.32$ Mb and $\sigma_1 = 9.34$ Mb[32], respectively. Since the ionization threshold for the process Ne²⁺ → Ne³⁺ (~63 eV) is well above the photon energy (50 eV), it is justified to neglect any further loss channel in Eq. (6) and thus, in principle, all population can be aggregated in Ne²⁺.

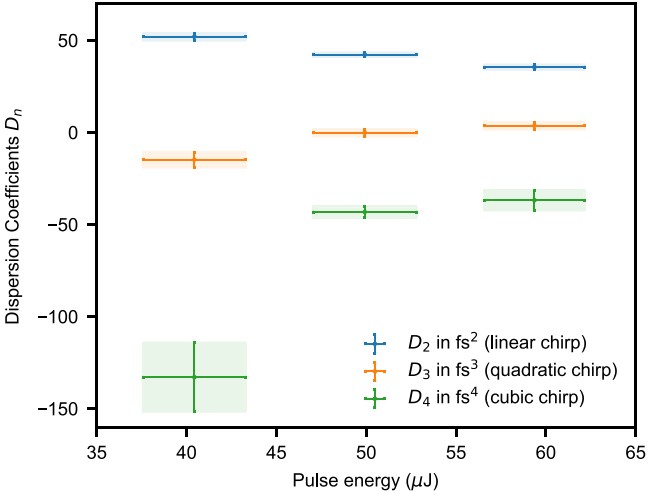

**Fig. 4 Frequency-chirp characterization, analysis of dependence on the FEL pulse energy.** Measured dispersion coefficients (blue dots: quadratic order; orange dots: cubic order; green dots: quartic order) for different FEL-pulse energy ranges between 35 and 45 μJ, 45 and 55 μJ, and 55 and 65 μJ, respectively. Given values of pulse energy represent the mean over ~5000 FEL shots, respectively, and quoted errors are standard deviations. Errors of dispersion coefficients specify best polynomial fit errors accounting for the errors due to error-function curve fitting to measured time-delay data. It should be noted that the stated numerical values for the different orders of chirp, in units of $fs^2$, $fs^3$, and $fs^4$, are displayed on the same scale on the vertical axis, which does not reflect their relative contribution to the total nonlinear chirp (see "Supplementary Note 2, Order-of-magnitude estimation of nonlinear chirp contributions").

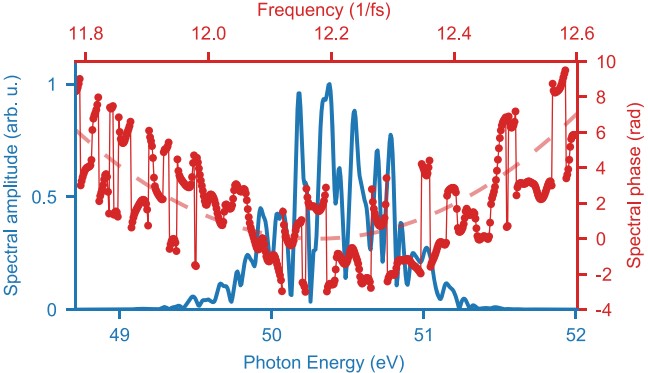

**Fig. 5 The effect of dispersion on partially coherent FEL fields.** Simulated spectral phase and amplitude of a 50.27-eV center-photon energy FEL pulse with a 50-fs temporal duration and a coherence length of 3 fs obtained from the model introduced in ref. [24]. The parabolic phase offset (indicated by the red dashed line) originates from an imposed linear chirp of $D_2 = 2\ fs^2$.

In order to model the temporal intensity profile of pump (or gate) pulses, $I_{pu}(t) \propto |E_{pu}(t)|^2$, we employ realistic stochastic SASE-FEL pulses[24] and account for a continuous frequency modulation (chirp). This approach to model SASE-FEL pulses builds on "colored" noise and incorporates partial coherence by setting temporal and spectral boundaries in accordance with the averaged temporal FEL-pulse duration and spectral bandwidth known from the experiment. The imparted spectral phase $\Phi(\omega)$ on the simulated initial partially coherent fields include the dispersion coefficients $D_n$ according to Eq. (2). Effectively, the frequency-domain electric field can be written as

$$\tilde{E}_{FEL}(\omega) = S(\omega)\exp\{i[\varphi_{SASE}(\omega) + \Phi(\omega)]\},$$

where $\varphi_{SASE}(\omega)$ is the partially coherent (i.e., not fully random) phase given by numbers between $-\pi$ and $\pi$ at all frequencies and $S(\omega)$ is the spectral amplitude. Hereby, $\varphi_{SASE}(\omega)$ varies from shot to shot, while $\Phi(\omega)$ remains constant for each particular chirp setting across the SASE pulse ensemble. The resulting total spectral phase and the spectrum for a single calculated pulse are illustrated in Fig. 5.

While each individual simulated pulse has a noise-like temporal structure consisting of a train of intensity bursts, averaging over an ensemble of several pulses, we obtain a Gaussian-shaped pulse envelope with $4 \times 10^{13}\ Wcm^{-2}$ peak intensity and 50-fs FWHM pulse duration. These assumptions are commensurate with currently available pulse parameters at the XUV free-electron laser in Hamburg (FLASH) and correspond to on-target pulse energies on the order of 10 μJ in a ~25-μm diameter beam focus[33]. For these conditions, the time-dependent fractional ion population is calculated and presented in Fig. 1b–d, predicting a highly abundant final $Ne^{2+}$ population on the order of ~70%. In the focused beam (assuming a small target-gas volume compared to the Rayleigh length), this forms an almost transparent medium for 50-eV photons. It should be noted that both the temporal pulse shape and its duration are sensitive to the spectral phase and may be disturbed for large values of nonlinear dispersion coefficients $D_n$ [cf. Eq. (2)], while the rate-equation-based sequential absorption of single XUV photons is independent of a possible chirp of the FEL pulses. Thus, the gate is formed through the phase-insensitive nonresonant ionization during the pump pulse, and Eqs. (4)–(6) hence capture well the time-dependent relative abundancies of the different neon ionic species.

Given the time-dependent populations of the different charge states, we now calculate the photo-absorption of the SASE-FEL probe pulses in the neutral/ionic neon target-gas mixture. For this calculation, we assume that the electric fields of the probe pulses, $E_{pr}(t - \tau)$, are identical time-($\tau$)-delayed copies of the pump pulses,

$E_{pu}(t)$. Considering the instantaneous response of the time-dependent continuum ionization dipole moment $d_n(t) = \frac{1}{2}i\pi|d_{n,gE}|^2 E_{pr}(t)N_n(t)$ for each charge state $n$, we can describe the time-delay-dependent absorbance[34] in terms of ODs as

$$A(\tau, \omega) \propto \sum_{n=0}^{1} \sigma_n \cdot \Im\left\{i\frac{\mathcal{F}[E_{pr}(t - \tau) \cdot N_n(t)]}{\mathcal{F}[E_{pr}(t)]}\right\}, \quad (7)$$

where we have factored out the squared ground—continuum matrix elements $|d_{n,gE}|^2$ in the numerator, whereby $\mathcal{F}$ denotes the Fourier transform and $\Im$ the imaginary part. The summation in Eq. (7) only includes the neutral and singly excited charge states, i.e., $n = 0, 1$, since the single-photon ionization loss out of $Ne^{2+}$ can be safely neglected, i.e., $\sigma_2 = 0$ for $Ne^{2+} \rightarrow Ne^{3+}$. Each fractional neutral/ionic contribution to the total absorbance is weighted by its respective population yield $N_n(t)$, which is determined by the pump pulse $I_{pu}(t)$ centered at $t = 0$ according to the solution of Eqs. (4)–(6).

**Experimental beamline setup.** The setup includes an XUV autocorrelator[25] in order to geometrically split the FEL beam into XUV-pump and delayed XUV-probe pulses (incremental time-delay step size: 10 fs), and the ~25-μm diameter (FWHM) focus at the user beamline BL2[35] at FLASH. The beam is focused into a neon-filled gas cell with moderate atomic target density on the order of $10^{18}\ cm^{-3}$. The 2-mm inner diameter of the cell is an order of magnitude smaller than the Rayleigh length, which ensures that the entire interaction volume lies within the focal peak-intensity region. Behind the cell, the transmitted radiation is spectrally dispersed using a variable-line-spacing grating and detected with an XUV-sensitive CCD camera. At the camera position, in the beam's far field, the pump and probe pulses do not spatially overlap and the transmitted spectrum of the probe pulse $I_{pr}(\omega)$ can be separately recorded. The incoming FEL photon spectra before passing through the target, termed reference spectra $I_{ref}(\omega)$, were recorded for each single pulse using the online variable-line-spacing grating spectrometer at FLASH[26]. The FEL pulse energy is measured using the non-invasive shot-to-shot gas monitor detector[35] installed in front of the experimental setup. Assuming on average 50-fs-duration Gaussian pulses and a beamline transmission of ~25%, we estimate the on-target peak intensity to reach the mid-$10^{13}$-$Wcm^{-2}$ range.

**Evaluation of the relative optical density (rel. OD).** In order to eliminate slight systematic spectral discrepancies between probe and reference spectra due to spatial inhomogeneities in the frequency distribution across the geometrically split beam profile, we evaluate the absorbance in units of the rel. OD, that is, the OD after subtracting the mean value of $A_{exp}(\tau, \omega)$ along the vertical ($\tau$) dimension of the data set. Effectively, this cancels out a frequency offset, while all $\tau$-dependent absorption changes remain. By no means does this affect the mathematical model used to fit the data from which the dispersion coefficients are retrieved.

## Data availability
The data that support the findings of this study are available from the corresponding authors upon reasonable request.

## Code availability
The computer codes used for model calculations are available from the corresponding authors upon reasonable request.

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

## Acknowledgements

We gratefully acknowledge technical support from C. Kaiser and B. Knape. We gratefully acknowledge the technical and scientific teams at FLASH, in particular Dr G. Brenner, for their support during the experiment. We acknowledge funding from the European Research Council (ERC) (X-MuSiC 616783). Z.-H.L. acknowledges the support of the Singapore Ministry of Education (RG105/17 and MOE2018-T2-1-081). H.Z. acknowledges the support of the BMBF (Project No. 05K13PM2).

## Author contributions

T.D., C.O., and T.P. conceived the XUV-FEL chirp characterization method and M.R., L.A., M.H. contributed to the design of the experiment. T.D., M.R., L.A., M.H., A.M., D.W. set up the experiment and performed data collection together with V.S., P.B., G.D.B., C.C.C., Y.M., A.R.A., T.G., and Z.-H.L., while S.R., M.B., and H.Z. provided the split and delay unit at FLASH, and P.R. contributed in discussing the results. S.D. and R.T. carried out interfacing and optimization of the experiment with FLASH. A.E. set up the accelerator configuration. T.D. performed data analysis and model simulations. S.M.C. calculated ionization cross sections. T.D., C.O., and T.P. wrote the paper. All authors discussed the results and contributed to the final manuscript.

## Funding

## Competing interests

The authors declare no competing interests.
