## [Peer Review File · Nature Communications]

Reviewers' Comments:

Reviewer #1:

Remarks to the Author:

This is an interesting instrumental development which is suitable for publication in Nature Communications. The authors have developed a method for measuring the average chirp of pulses from a Free-Electron Laser, and as they note, this field of science is developing very quickly and requires ever more sophisticated diagnostics and tools for characterisation of the optical properties. This renders the paper important and timely.

I have only comments which the authors may optionally wish to consider.

SASE FELs show large fluctuations in the properties of the light emitted. The present results include measurements sorted according to pulse energy, demonstrating that the chirp changes significantly with pulse energy. Have measurements been done under nominally the same conditions of pulse duration and energy, but with different accelerator tunes, e.g. at widely different times? This would be useful for indicating to users whether they have to measure the chirp every time it is an important parameter, or whether it can be reliably predicted.

It is difficult to understand some details of the experimental arrangement until getting to the Methods section. This is perhaps journal style, but I think a couple of sentences in the main body would help. In the "Experimental realization" section, it would be helpful if reference 6 of the Supplementary materials was added. Somewhere in this section, it would be useful to say that Iref and Ipr spectra are measured simultaneously by the detector as they are spatially separated. Otherwise the reader is puzzled by the statement "Their pulse spectra are measured separately on a single-shot basis", although fig. 1 shows only a single schematic spectrum. The mystery is resolved in the Methods section, but it would help if it is mentioned earlier. Also in the Methods section, it would be helpful to have the time resolution of the autocorrelator stated.

The authors call the method a "temporal knife-edge" although it is a blunt knife because the time resolution is of the order of the pulse duration, 50 fs. Nevertheless, it is sharp enough.

It is also a gate that opens, but does not close quickly. At the very end, the authors mention the application to hard x-rays, citing ref 24. It will be interesting to see the method applied using core ionization, as in this case the gate can shut quickly by Auger decay.

K. C. Prince

Reviewer #2:

Remarks to the Author:

I have been asked to review a manuscript submitted by Ding et al. entitled "Measuring the frequency chirp of extreme-ultraviolet free electron laser pulses by transient absorption spectroscopy." This manuscript presents an interesting measurement made at an x-ray free electron laser (XFEL) facility. Using XUV transient absorption spectroscopy the authors are able to extract a time-energy correlation of an aggregate XUV pulse.

As the Author's point out, most XFEL facilities operate on the principle of self amplification of spontaneous emission (SASE) which builds up from noise. Therefore every pulse from a SASE XFEL is different. However the measurement presented in the manuscript is averaged over a large number of pulses, while details of any individual pulse is lost, the Authors are able to extract an underlying time-energy relationship which is common to all the X-ray pulses accumulated. The introduction and abstract call this quantity the frequency chirp of the XUV pulse, but I am not sure that is the correct

terminology (see comments below). The time-energy structure observed is most certainly due to a time-energy correlation of the electron bunch generating the X-ray radiation. So in my opinion, what this work actually demonstrates is a way to measure this time-energy correlation in the electron bunch.

Even though the manuscript is very thorough in describing the transient absorption scheme, after reading the manuscript and the supplementary material I was left with a number of unanswered questions:

- The measured chirp could be due to space-charge effects as the Authors suggest, or it could just mean that the bunch was not fully compressed in the final bunch compressor of the accelerator. The Authors should specify how the machine was configured, was the bunch fully compressed? I would guess that the XFEL would have better performance if the electron bunch is left slightly over (or under) compressed to mitigate some space charge effects. Perhaps this slight offset from the maximum compression is actually what the Authors measure?

- To this previous point, for the description of a measurement technique, very few details were given about the machine parameters: what was the electron bunch charge? What about the bunch length? Was the bunch fully compressed? What was the peak current in the final bunch compressor? What was the electron beam energy?

+ I suspect that the measured D2 would have a strong correlation with the peak current in the final electron bunch compressor. It would be quite interesting if the authors made an additional sub-panel for Figure 4 that plots the extracted spectral phase coefficients as a function of peak current (in addition to pulse energy).

- The introduction and abstract refer to the measured quantity as the frequency chirp of the XUV pulse. This nomenclature seems very confusing to me. A single pulse from the XFEL is made up of a train of incoherent SASE bursts. The spectral phase of this pulse train will be quite complicated, and I do not believe that the proper description of the spectral phase of a SASE pulse is given by Eqns. 1 or 2. In fact, I find these equations misleading.

Instead, I think the Author's have done a very nice job to graphically describe the measured quantity in panel a of Figure 1. What the Authors are actually observing is that the different SASE bursts in the XUV pulse train have different colors, which change systematically in time, and result from a time-energy correlation in the electron bunch.

In my opinion, this seems more similar to the 'femtochirp' observed in high harmonic generation [J. Mod. Opt. 52, 379-394 (2005)], which is not well described by Eqn. 1. Perhaps this analogy is not exactly correct because the femtochirp is associated with a deviation from a regular burst spacing often observed in HHG. However the chirp measured in this experiment is on the timescale associated with the electron bunch length and not the XFEL coherence length. I believe that Eqn. 1 is a better description of the chirp associated with the XFEL coherence length, which is not measured in this experiment.

+ Related to this point, it is not clear how the Authors simulate the SASE pulse. Do they actually use a train of nearly transform-limited, incoherent Gaussian pulses in the simulation, or do they use a single Gaussian envelope with variable width? From the description given in the model simulation section, it sounds like the latter, which is a very poor model for SASE radiation. The Authors should clarify this description, and if they have not done so, repeat the model simulations with a more realistic SASE pulse.

Given the number of unanswered questions I cannot recommend publication of this article in its current form. Moreover, while I find the transient absorption measurement quite interesting I am not convinced that this measurement will be useful at most XFEL facilities. Even if the Authors address the concerns I have raised I still do not think that this work will meet the broader impact metric expected for publication in Nature Communications. Extracting the time-energy correlation of the electron bunch can be done in a number of ways, for example the same information could be obtained from an X-band radio frequency transverse deflector cavity [Nat. Comm. 5 3762 (2014)]. I suspect that most of the "tuning" information produced by this measurement could be obtained through simple measurement of the peak current in the electron bunch compressor. I am not sure that the XFEL community will find much utility in this technique. Rather I believe that future XUV-pump/XUV-probe transient absorption measurements will have to understand this effect, and be sure to quantify it before interpreting any measurements.

Therefore, I do not recommend publication of this work in Nature Communication. However, I think that a revised version of this manuscript would do very well in a more specialized journal.

Reviewer #3:

Remarks to the Author:

This paper presents a new technique for the direct measurement of the frequency chirp in short pulses at EUV wavelengths. The authors use a pump and probe technique where the pump pulse gates the absorption of the probe that is then spectrally analysed. This allows to correlate the detected frequency spectrum to the delay between pump and probe and therefore to correlate the measured spectrum to its arrival time, i.e. its position along the pulse.

The method applied to a well behaving, quasi coherent pulse, should lead to a complete reconstruction of the field amplitude and phase, as in FROG implemented in ultrashort pulses emitted by solid state lasers. The absolute determination of the field properties in terms of amplitude and phase is an important achievement and the method proposed based on optical gating is promising and worth of consideration. The paper is well written and should be published, after the authors have considered the following remarks:

The experimental test of the method has been carried out on a SASE FEL and SASE pulses are not well behaving pulses. They are constituted by an ensemble of sub-pulses with no relative phase correlation between them. Therefore, the method applied to a SASE source can be used to retrieve an average spectral phase that is useful to reconstruct the average frequency chirp. In averaged measurements the phase remains undetermined, as it changes randomly from one coherence region another within a single pulse, and with a distribution that also changes from one pulse another. In SASE, the structure of the pump pulse, which is a replica of the probe, is equally not well behaving, and the reconstruction of the phase could still be non trivial, even if a single shot measurement of a spectrogram would be available. Conversely, an absolute phase dependence vs the longitudinal coordinate along the bunch should be possible for "well" behaving pulses. The method could provide the additional information of the pulse phase of well behaved pulses, as those generated in a seeded FEL. A remark on these aspects should be added to the manuscript.

The coherence length of a SASE FEL pulse depends on the ρ FEL parameter as $L_c = 1/(2k\rho)$ where k is the radiation central k vector. The quoted FWHM width of 1 eV is the result of a combination of chirp and intrinsic FEL process bandwidth. The manuscript doesn't provide the elements to understand what is the expected cooperation length the gain length, and how these parameters fit with the measured spectra. The method section should provide a sample of measured spectra and a measurement of the beam longitudinal phase space, if possible. This simple measurement would strengthen the claim, if e.g. the beam energy chirp would match the observed dependence of the resonance versus delay.

The gating process is based on the depletion of Ne and Ne⁺ ions. The ionisation is not only induced by the pump pulse, but by the combination of pump and probe.

- What is the energy balance between pump and probe ?

- In the calculation based on rate equations presented in the supplementary material, it seems only the pump field is considered. What is the expected ionisation effect of the probe only ?

-Some substantial changes in the dispersion coefficients vs the fluence was observed. Such a dependence could introduce some arbitrariness in the output chirp coefficients, and should be further commented. No explanation is given e.g. about the important change especially of D4 at moderately low fluence.

Luca Giannessi

Responses to reviewers' remarks

Please find below our detailed reply to the reviewer reports and the corresponding changes we made in the manuscript. The corresponding remarks of the reviewers are inserted in green; the passages we deleted from our manuscript are marked in red, while the additional text we inserted or changed in our manuscript is marked in blue.

All changes to manuscript are further colour highlighted in the annotated manuscript text file, also including minor rephrasing.

Reviewer #1 (Remarks to the Author):

This is an interesting instrumental development which is suitable for publication in Nature Communications. The authors have developed a method for measuring the average chirp of pulses from a Free-Electron Laser, and as they note, this field of science is developing very quickly and requires ever more sophisticated diagnostics and tools for characterisation of the optical properties. This renders the paper important and timely.

Thank you for your positive report and the valuable suggestions, which helped us to further strengthen the paper and to improve the presentation.

I have only comments which the authors may optionally wish to consider.

SASE FELs show large fluctuations in the properties of the light emitted. The present results include measurements sorted according to pulse energy, demonstrating that the chirp changes significantly with pulse energy. Have measurements been done under nominally the same conditions of pulse duration and energy, but with different accelerator tunes, e.g. at widely different times? This would be useful for indicating to users whether they have to measure the chirp every time it is an important parameter, or whether it can be reliably predicted.

Unfortunately, in these first experiments the accelerator settings have not been altered throughout the measurement. The suggested idea of different accelerator tunes and settings on the chirp is certainly an interesting avenue for future exploration but would require a dedicated new beamtime, so we would like to stick to the presented dataset for the first proof-of-principle demonstration of this new method.

It is difficult to understand some details of the experimental arrangement until getting to the Methods section. This is perhaps journal style, but I think a couple of sentences in the main body would help.

In the “Experimental realization” section, it would be helpful if reference 6 of the Supplementary materials was added.

- 1) In the manuscript on page 9, we added Ref. 25 [Wöstmann, M. *et al.* The XUV split-and-delay unit at beamline BL2 at FLASH. *J. Phys. B At. Mol. Opt. Phys.* **46**, 164005 (2013)] and change the first sentence to now read:

“The beamline setup includes an XUV autocorrelator²⁵ and allows for the preparation of XUV-pump and time-delayed XUV-probe FEL pulses, [...]”

Somewhere in this section, it would be useful to say that Iref and Ipr spectra are measured simultaneously by the detector as they are spatially separated. Otherwise the reader is puzzled

by the statement “Their pulse spectra are measured separately on a single-shot basis”, although fig. 1 shows only a single schematic spectrum. The mystery is resolved in the Methods section, but it would help if it is mentioned earlier.

- 2) Following this suggestion, in the manuscript on page 9, we deleted the sentence:

Their pulse spectra are measured separately on a single-shot basis

and replaced it by:

While the pump and probe pulses spatially overlap within the focal interaction volume, they are again separated in the far field behind the focus and the transmitted spectrum of the probe pulse, denoted by $I_{\text{pr}}(\omega)$, can be measured separately. Spectra taken before passing through the neon target are termed reference spectra, $I_{\text{ref}}(\omega)$, and were measured simultaneously using the parasitic online photon spectrometer at FLASH²⁶.

[26] Brenner, G. et al. First results from the online variable line spacing grating spectrometer at FLASH. Nucl. Instruments Methods Phys. Res. Sect. A Accel. Spectrometers, Detect. Assoc. Equip. **635**, S99–S103 (2011).

Also in the Methods section, it would be helpful to have the time resolution of the autocorrelator stated.

- 3) We added this information on the time resolution (autocorrelator step size) in the supplementary material on page 5, sentence in parentheses:

“The setup includes an XUV autocorrelator⁸ in order to geometrically split the FEL beam into XUV-pump and delayed XUV-probe pulses (incremental time-delay step size: 10 fs), [...]”

The authors call the method a “temporal knife-edge” although it is a blunt knife because the time resolution is of the order of the pulse duration, 50 fs. Nevertheless, it is sharp enough.

- 4) Yes, the knife-edge is sufficiently “sharp” if the duration of the gate (pump-pulse duration) is on the order of the probe-pulse duration. We now comment on this in the manuscript on page 4, by adding the text in parentheses:

If the target depletion time is sufficiently short (i.e., on the order of the probe-pulse duration), the step-like transmission gate function serves as a “temporal knife-edge”^{16,17} [...].

It is also a gate that opens, but does not close quickly. At the very end, the authors mention the application to hard x-rays, citing ref 24. It will be interesting to see the method applied using core ionization, as in this case the gate can shut quickly by Auger decay.

K. C. Prince

- 5) Thank you for raising this interesting point. We follow this suggestion and now discuss this point in the manuscript, page 12, second paragraph, by adding the sentence: In hard-x-ray absorption spectroscopy and when applied specifically to core electrons, a second (intrinsic) shutter is at work due to the Auger effect and the ultrafast refilling of the core hole, which may shut the open gate (transparency gate

through core ionization) again quickly on a sub-pulse-duration timescale and may be exploited in the future to further optimize the method.

Reviewer #2 (Remarks to the Author):

I have been asked to review a manuscript submitted by Ding et al. entitled “Measuring the frequency chirp of extreme-ultraviolet free electron laser pulses by transient absorption spectroscopy.” This manuscript presents an interesting measurement made at an x-ray free electron laser (XFEL) facility. Using XUV transient absorption spectroscopy the authors are able to extract a time-energy correlation of an aggregate XUV pulse.

As the Author’s point out, most XFEL facilities operate on the principle of self amplification of spontaneous emission (SASE) which builds up from noise. Therefore every pulse from a SASE XFEL is different. However the measurement presented in the manuscript is averaged over a large number of pulses, while details of any individual pulse is lost, the Authors are able to extract an underlying time-energy relationship which is common to all the X-ray pulses accumulated. The introduction and abstract call this quantity the frequency chirp of the XUV pulse, but I am not sure that is the correct terminology (see comments below). The time-energy structure observed is most certainly due to a time-energy correlation of the electron bunch generating the X-ray radiation. So in my opinion, what this work actually demonstrates is a way to measure this time-energy correlation in the electron bunch.

Thank you for your critical report and for considering our work interesting.

We fully agree that the extracted averaged time-energy relationship of the SASE XFEL radiation can be related to a systematic time-energy correlation of the electron bunch. In this regard (see point “1”) below), our revised paper now contains a clarifying section about the FEL machine settings, the impact of the compression settings and space-charge effects on the FEL-frequency chirp. Furthermore, we fully resolve the meaning of the terminology “frequency chirp” for SASE XUV pulses with our explanations below (see points “2”) and “3”) with according changes in the manuscript.

We would, however, also like to clearly emphasize that it has neither been our intention to measure the time-energy correlation of the electron bunch, nor to explore a close connection between the electron bunch and the FEL light pulse. Our here introduced technique enables to measure and quantify the spectro-temporal properties of the FEL radiation directly (i.e. the XUV output), *in-situ*, and at the point of interaction with the target of a user experiment. So, by no means is it our objective to provide an (alternative, indirect) way to measure the time-energy correlation in the electron bunch, even though our approach may provide the opportunity for further benchmarking photon-pulse diagnostics derived from electron-beam measurements and to test the impact of SASE-tuning parameters on the photon-pulse properties.

Furthermore, in a typical user experiment it is mainly the average ensemble FEL properties that determine the outcome and have a key influence on the investigated processes, since often counting statistics need to be acquired over several single-shot events (e.g., scanning pump-probe time delays) which feeds into understanding the physical process. Therefore, also averaged FEL on-target photon properties are a highly valuable input quantity for the large and diverse community of FEL users.

In summary, what makes our work unique and interesting to a wide scientific audience across sub-field boundaries is the *direct* and *quantitative* access to the spectro-temporal properties of SASE FEL light pulses without relying on any additional external field, and measured *in situ* at the point of the user experiment.

Even though the manuscript is very thorough in describing the transient absorption scheme, after reading the manuscript and the supplementary material I was left with a number of unanswered questions:

- The measured chirp could be due to space-charge effects as the Authors suggest, or it could just mean that the bunch was not fully compressed in the final bunch compressor of the accelerator. The Authors should specify how the machine was configured, was the bunch fully compressed? I would guess that the XFEL would have better performance if the electron bunch is left slightly over (or under) compressed to mitigate some space charge effects. Perhaps this slight offset from the maximum compression is actually what the Authors measure?
- To this previous point, for the description of a measurement technique, very few details were given about the machine parameters: what was the electron bunch charge? What about the bunch length? Was the bunch fully compressed? What was the peak current in the final bunch compressor? What was the electron beam energy?

Indeed, looking at the accelerator settings and diagnostics of the electron bunch reveals that it was not fully compressed. This seemingly unusual operation mode has actually been set on purpose, with the goal to generate a broader SASE photon-energy bandwidth, the latter being useful for broadband transient-absorption measurements. Directly considering the electron-bunch compression setting is an interesting point and we thank the Reviewer for drawing this connection. Its discussion helps to even further improve the presentation of the experiment. We thus added a section about the machine configuration to the supplementary material, answering all questions on the electron-bunch parameters, and also added its main conclusion to the main text, namely that the electron bunch was not fully compressed in order to achieve broadband photon bandwidth for our transient absorption measurements:

1) In the manuscript on page 9, we added:

The accelerator was operated at a moderate compression, chirping the electron bunch (for details, see Methods, Accelerator settings and performance) in order to provide a broad XUV photon bandwidth for transient-absorption measurements.

Supplementary material (Methods), pages 4-5:

Additional section: Accelerator settings and performance

The measurements have been performed at the free-electron laser in Hamburg, FLASH⁶. With the accelerator settings of 1.2-kA peak current, an emittance of 1 mm-mrad, an electron energy spread of $\Delta E = 0.2$ MeV at 515 MeV electron beam energy, and a beta function of 10 m, we get a (natural) photon spectral bandwidth of about $(\Delta\omega/\omega)_{\text{FEL}} = 0.4\%$ (private communication with M.V. Yurkov). In fact, the spectral bandwidth is a slow function of the electron beam parameters. Based on the experimentally measured FEL spectra (see Fig. 3a in the main text), the XUV spectral bandwidth, $\Delta\omega/\omega = 1.6\%$, was significantly larger than the natural bandwidth, there is strong evidence of a chirp. This finding is supported by the electron beam compression settings that were not set to full compression, which was done on purpose in order to achieve a broader photon bandwidth for transient absorption measurements. Unfortunately, no dispersive electron phase-space measurements were performed to quantify the RF-induced chirp of the electron bunch.

In addition, longitudinal space-charge fields, induced by a current spike, can induce an energy chirp on the electron bunch⁷. However, since the measured bunch current profile [transverse deflecting RF structure (TDS) measurement with a 120-fs FWHM electron bunch, see Fig. S2] does not show a current spike, the influence of

such space-charge effects seems low. We therefore assume that the broad-bandwidth accelerator tuning, utilizing a not fully compressed electron bunch, is the main explanation for the observation of a significant chirp of the FEL photons which is discussed in the main text.

[7] Dohlus et al, Nuclear Instruments and Methods in Physics Research A **530**, 217 (2004)

Fig. S2: TDS measurement of the electron-bunch profile (green line). The reconstructed width (FWHM) is about 120 fs.

This additional section is based on a discussion with the SASE-tuning expert Arvid Eislage who has set up the machine configuration for this specific experiment and who we now include in the list of authors.

+ I suspect that the measured D2 would have a strong correlation with the peak current in the final electron bunch compressor. It would be quite interesting if the authors made an additional sub-panel for Figure 4 that plots the extracted spectral phase coefficients as a function of peak current (in addition to pulse energy).

Unfortunately, the peak current information of the electron bunch is not accessible when driving the SASE process of generating XUV photons. The XUV pulse energy however can be parasitically measured with the available photon diagnostics. Thus, we cannot directly relate the XUV spectral phase coefficients to the peak current of the lasing electron bunch. However, we are confident that with the added section on the machine settings (see above), revealing a not fully compressed electron bunch, an explanation of the connection between electron bunch compression and XUV chirp is given. It is clearly beyond the scope of the reported work to perform in-depth correlation studies between the input electron bunch and XUV photon output, but this is an interesting route for future studies of FEL machine performance: exploring and

benchmarking novel FEL operational regimes, e.g. to generate broadband XUV photon output for attosecond-pulse generation by means of the electron bunch compressor setting, aided by the presented direct spectro-temporal measurement of the FEL light pulses.

- The introduction and abstract refer to the measured quantity as the frequency chirp of the XUV pulse. This nomenclature seems very confusing to me. A single pulse from the XFEL is made up of a train of incoherent SASE bursts. The spectral phase of this pulse train will be quite complicated, and I do not believe that the proper description of the spectral phase of a SASE pulse is given by Eqns. 1 or 2. In fact, I find these equations misleading.

- 2) By eqns. 1 and 2 we refer to the frequency chirp of an ensemble of SASE pulses (also see our explanation in points “3”- “6”) below). In order to avoid confusions by future readers, on page 5 of the main text we deleted the passage **Mathematically, for a chirped pulse, the time delay versus frequency relationship follows the condition** and replaced it by **Hereby the statistical ensemble of SASE FEL pulses can be modelled with the partial-coherence method²⁴ which assumes a fluctuating spectral phase, on top of which an average frequency chirp can be added (see Methods, Computational model and Fig. S1 for details). Only considering this average chirp, the time delay versus frequency relationship is mathematically expressed by [...].**

[24] Pfeifer, T., Jiang, Y., Düsterer, S., Moshhammer, R. & Ullrich, J. Partial-coherence method to model experimental free-electron laser pulse statistics. *Opt. Lett.* **35**, 3441–3 (2010).

Instead, I think the Author’s have done a very nice job to graphically describe the measured quantity in panel a of Figure 1. What the Authors are actually observing is that the different SASE bursts in the XUV pulse train have different colors, which change systematically in time, and result from a time-energy correlation in the electron bunch.

In my opinion, this seems more similar to the ‘femtochirp’ observed in high harmonic generation [J. Mod. Opt. 52, 379–394 (2005)], which is not well described by Eqn. 1. Perhaps this analogy is not exactly correct because the femtochirp is associated with a deviation from a regular burst spacing often observed in HHG. However the chirp measured in this experiment is on the timescale associated with the electron bunch length and not the XFEL coherence length. I believe that Eqn. 1 is a better description of the chirp associated with the XFEL coherence length, which is not measured in this experiment.

+ Related to this point, it is not clear how the Authors simulate the SASE pulse. Do they actually use a train of nearly transform-limited, incoherent Gaussian pulses in the simulation, or do they use a single Gaussian envelope with variable width? From the description given in the model simulation section, it sounds like the latter, which is a very poor model for SASE radiation. The Authors should clarify this description, and if they have not done so, repeat the model simulations with a more realistic SASE pulse.

The partial-coherence method [Pfeifer et al., *Opt. Lett.* **35**, 3441–3 (2010)] used in this work is a stochastic approach to model pulses that mimic the characteristics of SASE pulses with pulse-to-pulse fluctuations both in amplitude and phase. Eqns. 1 and 2 describe only the systematic part of the ensemble average over such pulses. In some sense it may indeed be understood as the ‘femtochirp’ in HHG (i.e., the chirp of a single harmonic), whereas in HHG also the underlying ‘attochirp’ is roughly constant from shot to shot, while for the case of SASE pulses the analogous quantity statistically fluctuates from shot to shot. Therefore, we think drawing a closer correspondence to HHG may not help a reader too much. Instead we think it is more

instructive to illustrate this process directly with modeled SASE pulses following the partial-coherence method. We now clarify this by

- 3) adding Figure S1 in the supplementary material, page 2, which illustrates the simulated partially-coherent spectral phase and amplitude structure of an exemplary single SASE pulse that is chirped.

Fig. S1: Simulated spectral phase and amplitude of a 50.27-eV center-photon energy FEL pulse with a 50-fs temporal duration and a coherence length of 3 fs obtained from the model introduced in Ref. ³. The parabolic phase offset (indicated by the red dashed line) originates from an imposed linear chirp of $D_2 = 2 \text{ fs}^2$.

- 4) Related to this point, in the supplementary material, page 1, paragraph 2, we added the passages:
This approach to model SASE-FEL pulses builds on “colored” noise and incorporates partial coherence by setting temporal and spectral boundaries in accordance with the averaged temporal FEL-pulse duration and spectral bandwidths known from the experiment.
- 5) and (supplementary material, page 2, paragraphs 1-2):
“Effectively, the frequency-domain electric field can be written as $\tilde{E}_{\text{FEL}}(\omega) = S(\omega) \exp\{i[\varphi_{\text{SASE}}(\omega) + \Phi(\omega)]\}$, where $\varphi_{\text{SASE}}(\omega)$ is the partially coherent (i.e., not fully random) phase given by numbers between $-\pi$ and π at all frequencies and $S(\omega)$ is the spectral amplitude. Hereby, $\varphi_{\text{SASE}}(\omega)$ varies from shot to shot, while $\Phi(\omega)$ remains constant for each particular chirp setting across the SASE pulse ensemble. The resulting total spectral phase and the spectrum for a single calculated pulse are illustrated in Fig. S1.

While each individual simulated pulse has a noise-like temporal structure consisting of a train of intensity bursts, averaging over an ensemble of several pulses, we obtain a Gaussian-shaped pulse envelope with $4 \times 10^{13} \text{ Wcm}^{-2}$ peak intensity and 50 fs FWHM pulse duration.”

- 6) In the main text, page 7, we refer to this additional information on the computational method in the supplement by inserting the sentence:

The modeled stochastic pulses vary from shot to shot in their spiky sub-structure due to a random relative phase for each of them, while a continuous spectral-phase offset according to equation (2) (frequency chirping and modulation of the “macroscopic” pulse shape) is equally inherent to all pulses (cf. Methods, Fig. S1).

- 7) To avoid any unintentional connotation that we were to measure the spectral phase of each individual SASE pulse in the abstract, we replaced the wording **spectral phase** by **time-energy structure of FEL pulses**.

Given the number of unanswered questions I cannot recommend publication of this article in its current form. Moreover, while I find the transient absorption measurement quite interesting I am not convinced that this measurement will be useful at most XFEL facilities. Even if the Authors address the concerns I have raised I still do not think that this work will meet the broader impact metric expected for publication in Nature Communications. Extracting the time-energy correlation of the electron bunch can be done in a number of ways, for example the same information could be obtained from an X-band radio frequency transverse deflector cavity [Nat. Comm. 5 3762 (2014)]. I suspect that most of the “tuning” information produced by this measurement could be obtained through simple measurement of the peak current in the electron bunch compressor. I am not sure that the XFEL community will find much utility in this technique. Rather I believe that future XUV-pump/XUV-probe transient absorption measurements will have to understand this effect, and be sure to quantify it before interpreting any measurements.

Therefore, I do not recommend publication of this work in Nature Communication. However, I think that a revised version of this manuscript would do very well in a more specialized journal.

- 8) We thank the referee for pointing out the importance of the XTCAV installed at LCLS in the context of FEL-pulse-diagnostic methodologies. However, it should also be noted that the XTCAV measurement is performed on the electron bunch and thus information on the FEL radiation is only indirectly obtained from this measurement. We are happy to include this reference to the introduction part of our manuscript on page 3, having included the sentence:
A complementary route to diagnosing the temporal structure of SASE-radiation pulses is by measuring the FEL-lasing-induced energy losses in the electron bunch (for example with an X-band radio-frequency transverse deflecting cavity, XTCAV¹²), from which the temporal profile of the XUV/x-ray emission can be reconstructed.

[12] Behrens, C. *et al.* Few-femtosecond time-resolved measurements of X-ray free-electron lasers. *Nat. Commun.* **5**, 3762 (2014)

In order to further emphasize the novelty of our work we added the following sentence at the end of the introduction:

Since the reported technique provides on-target access to the spectro-temporal distribution of the XUV radiation, it is ideal for *in-situ* diagnosing of user experiments with sensitivity to the FEL-lasing performance. For instance, here, we experimentally observe a systematic dependence of the frequency chirp on the FEL pulse energy (increasing chirp for decreasing pulse energy).

- 9) Also, in order to further stress that our key objective is to characterize FEL light pulses (and to avoid emphasis on online FEL performance optimization), in the abstract, last sentence, we replaced the word **online** by **on-target measurement and**.

We hope that after the substantial improvements to the manuscript's content and presentation, and clarification of all points raised by the referee, s/he can join the other referees in recommending publication.

Being a direct spectro-temporal method for measuring *in-situ* (at the point of a user experiment) frequency chirps of an ensemble of FEL light pulses, without implementing additional fields for cross correlation, this new approach is opening doors in multiple directions and disciplines:

- *in-situ* optimization of FEL pulse duration by a spectrally sensitive measurement of the (average) arrival times directly of the FEL photons.
- precise measurements of time delays by future transient-absorption experiments at FELs in multiple research areas including atomic, molecular and condensed-matter dynamics as well as ultrafast materials science.
- advancing machine development by systematic studies of correlations between electron-bunch parameters (e.g. with high-level measurement through XTCAV) and photon pulse chirps (accessed by the here presented method). Besides its fundamental interest, this could also boost applications by exploring novel FEL operational regimes with broadband XUV/x-ray spectra for upcoming transient-absorption and multidimensional-spectroscopy nonlinear wave-mixing experiments at FELs.

Reviewer #3 (Remarks to the Author):

This paper presents a new technique for the direct measurement of the frequency chirp in short pulses at EUV wavelengths. The authors use a pump and probe technique where the pump pulse gates the absorption of the probe that is then spectrally analysed. This allows to correlate the detected frequency spectrum to the delay between pump and probe and therefore to correlate the measured spectrum to its arrival time, i.e. its position along the pulse.

The method applied to a well behaving, quasi coherent pulse, should lead to a complete reconstruction of the field amplitude and phase, as in FROG implemented in ultrashort pulses emitted by solid state lasers. The absolute determination of the field properties in terms of amplitude and phase is an important achievement and the method proposed based on optical gating is promising and worth of consideration. The paper is well written and should be published, after the authors have considered the following remarks:

First of all, thank you for your positive report and the valuable suggestions, which helped to further improve the presentation of this work and to discuss its potential for future applications also at seeded FELs.

The experimental test of the method has been carried out on a SASE FEL and SASE pulses are not well behaving pulses. They are constituted by an ensemble of sub-pulses with no relative phase correlation between them. Therefore, the method applied to a SASE source can be used to retrieve an average spectral phase that is useful to reconstruct the average frequency chirp. In averaged measurements the phase remains undetermined, as it changes randomly from one coherence region another within a single pulse, and with a distribution that also changes from one pulse another. In SASE, the structure of the pump pulse, which is a replica of the probe, is equally not well behaving, and the reconstruction of the phase could still be non trivial, even if a single shot measurement of a spectrogram would be available. Conversely, an absolute phase dependence vs the longitudinal coordinate along the bunch should be possible for "well" behaving pulses. The method could provide the additional information of the pulse phase of well behaved pulses, as those generated in a seeded FEL. A remark on these aspects should be added to the manuscript.

We fully agree that—implemented at a quasi-coherent seeded FEL and in combination with advanced analysis methods—the presented approach could provide the full phase and amplitude information. We now discuss this point and add the following passage

1) in the main text, on page 13 (discussion section):

In principle, it is possible to extract the complete phase and amplitude information of the XUV/x-ray FEL pulses from such transient-absorption spectrograms with an optical gate by performing advanced phase-retrieval calculations as recently demonstrated for mid-IR laser pulses²⁷. This, however, requires sufficiently well-behaved (phase and amplitude stable) pulses as, e.g., those generated in a seeded FEL²⁸, while the retrieval is expected to be particularly non-trivial for the case of SASE-FEL pulses, which are composed of temporally coherent sub-pulses with no relative phase correlation between them, and which fluctuate from shot to shot.

[27] Leblanc, A. *et al.* Phase-matching-free pulse retrieval based on transient absorption in solids. *Opt. Express* **27**, 28998 (2019).

[28] Allaria, E. *et al.* Highly coherent and stable pulses from the FERMI seeded free-electron laser in the extreme ultraviolet. *Nat. Photonics* **6**, 699–704 (2012).

The coherence length of a SASE FEL pulse depends on the ρ FEL parameter as $L_c = 1/(2k\rho)$ where k is the radiation central k vector. The quoted FWHM width of 1 eV is the result of a combination of chirp and intrinsic FEL process bandwidth. The manuscript doesn't provide the elements to understand what is the expected cooperation length the gain length, and how these parameters fit with the measured spectra. The method section should provide a sample of measured spectra and a measurement of the beam longitudinal phase space, if possible. This simple measurement would strengthen the claim, if e.g. the beam energy chirp would match the observed dependence of the resonance versus delay.

Thank you for pointing this out. Our revised manuscript now contains an additional section about the FEL machine parameters and the electron bunch compression setting. Unfortunately, no dispersive electron phase-space measurements were performed to quantify the RF-induced chirp. We can however produce an estimate of the intrinsic FEL bandwidth based on the electron bunch parameters, now included in the Methods section, which we further compare to the bandwidth of the experimentally measured FEL spectra reported in Fig. 3a of the main text.

2) In the manuscript on page 9, we added:

The accelerator was operated at a moderate compression, chirping the electron bunch (for details, see Methods, Accelerator settings and performance) in order to provide a broad XUV photon bandwidth for transient-absorption measurements.

Supplementary material (Methods), pages 4-5:

Additional section: Accelerator settings and performance

The measurements have been performed at the free-electron laser in Hamburg, FLASH⁶. With the accelerator settings of 1.2-kA peak current, an emittance of 1 mm-mrad, an electron energy spread of $\Delta E = 0.2$ MeV at 515 MeV electron beam energy, and a beta function of 10 m, we get a (natural) photon spectral bandwidth of about $(\Delta\omega/\omega)_{\text{FEL}} = 0.4\%$ (private communication with M.V. Yurkov). In fact, the spectral bandwidth is a slow function of the electron beam parameters. Based on the experimentally measured FEL spectra (see Fig. 3a in the main text), the XUV spectral bandwidth, $\Delta\omega/\omega = 1.6\%$, was significantly larger than the natural bandwidth, there is strong evidence of a chirp. This finding is supported by the electron beam compression settings that were not set to full compression, which was done on purpose in order to achieve a broader photon bandwidth for transient absorption measurements. Unfortunately, no dispersive electron phase-space measurements were performed to quantify the RF-induced chirp of the electron bunch.

In addition, longitudinal space-charge fields, induced by a current spike, can induce an energy chirp on the electron bunch⁷. However, since the measured bunch current profile [transverse deflecting RF structure (TDS) measurement with a 120-fs FWHM electron bunch, see Fig. S2] does not show a current spike, the influence of such space-charge effects seems low. We therefore assume that the broad-bandwidth accelerator tuning, utilizing a not fully compressed electron bunch, is the main explanation for the observation of a significant chirp of the FEL photons which is discussed in the main text.

[7] Dohlus et al, Nuclear Instruments and Methods in Physics Research A **530**, 217 (2004)

Fig. S2: TDS measurement of the electron-bunch profile (green line). The reconstructed width (FWHM) is about 120 fs.

Indeed, after a critical re-evaluation of the electron-bunch settings related to our measurement, it turns out that space-charge effects are playing only a minor role, and we now present evidence (i.e., our reply above concerning the machine settings and the absence of a current spike on the electron bunch), that the XUV chirp is mainly influenced by the not fully compressed electron bunch.

- 3) Accordingly, we revised our previous comments about space-charge effects in the manuscript on page 10, replacing the sentence:

The finding of a predominant linear chirp is consistent with indications from terahertz-field-driven streaking measurements¹⁰, and can be explained by space-charge effects of the lasing electron bunch¹.

by:

The finding of a predominant linear chirp is consistent with an energy chirp on the electron bunch which was not fully compressed, while collective space-charge effects are expected to play a minor role (see Methods, Accelerator settings and performance).

The gating process is based on the depletion of Ne and Ne⁺ ions. The ionisation is not only induced by the pump pulse, but by the combination of pump and probe.

- What is the energy balance between pump and probe ?

- 4) The pump vs. probe energy balance is about 2:1. It is certainly helpful to keep the probe-pulse energy lower than the one of the pump pulse in order to ensure a most direct interpretation of the measurement.

In the manuscript on page 10, we added:

We notice that the experiment was performed with a similar pump- vs. probe-pulse intensity sharing of about 2:1. In general, the pump pulse should be the more intense one such that it dominates the ionization process used for the gate.

- In the calculation based on rate equations presented in the supplementary material, it seems only the pump field is considered. What is the expected ionisation effect of the probe only ?

The ionization due to the less intense probe pulse only leads to a Ne^{2+} -ion abundance of about 30-40%. In combination with the more intense ionizing pump pulse, this ionization-offset effect mainly has an impact on the amplitude of the knife-edge (i.e., the erf-type transient-absorption trace), but it does not significantly affect its shape.

5) To clarify this in the manuscript, we thus added on page 11:

We do not explicitly treat the probe-induced ionization in the numerical modelling, as generally the probe pulse should be set to lower pulse energy than the pump pulse to ensure the latter to dominate the ionization dynamics of the gate. Even a slight ionization offset contribution by the probe pulse would mainly affect the contrast (i.e., amplitude) of the calculated OD-signal ramp, globally for all frequencies, without significantly modifying its shape and thus the extracted spectral-phase coefficients.

-Some substantial changes in the dispersion coefficients vs the fluence was observed. Such a dependence could introduce some arbitrariness in the output chirp coefficients, and should be further commented. No explanation is given e.g. about the important change especially of D_4 at moderately low fluence.

Luca Giannessi

Indeed, the D_4 chirp coefficient appears to change rather strongly compared to the other coefficients. We notice that Fig. 4 may give this implication of presenting the numerical values of different units (fs^2 , fs^3 and fs^4) shown on the same scale. With increasing spectral bandwidth (1.6% in our case) it thus becomes increasingly important to also observe higher orders of the spectral chirp to properly describe the average trend of the nonlinear spectral phase, while it is still clearly dominated by the lowest second-order linear chirp D_2 . A first estimation of the relative weight of the different Taylor expansion coefficients contributing to the actually measured dispersion function $\tau(\omega)$ can be given in terms of the absolute spectral bandwidth $\Delta\omega$. We believe that this estimation might be important for the reader in order to correctly assess the order of magnitude of the individual orders of chirp and their respective impact on the spectro-temporal pulse structure.

6) We thus add in the supplementary material, Methods, on pages 6-7:

The additional section: Order-of-magnitude estimation of non-linear chirp contributions

We approximate the measured dispersion function $\tau(\omega)$ [cf. equation (1) of the main text] by a third-order Taylor-series expansion about the center photon energy $\omega_L \approx 50.3$ eV. The measured absolute FEL spectral bandwidth at half-width at half maximum (HWHM), $\Delta\omega_{\text{HWHM}} = |\omega_{\text{min/max}} - \omega_L|$ (with the spectral wing positions ω_{min} and ω_{max}), is about 0.4 eV, or 0.6 rad/fs, and allows for an order-of-magnitude estimation of the individual Taylor expansion coefficients D_n of the spectral phase $\Phi(\omega)$ [cf. equation (2) of the main text]. Given the polynomial coefficients extracted

from the fit to the measured data for moderately low FEL-pulse energies between 35-45 μJ (see Fig. 4 of the main text), i.e., $D_2 = 52 \text{ fs}^2/\text{rad}$, $D_3 = -15 \text{ fs}^3/\text{rad}^2$, $D_4 = -133 \text{ fs}^4/\text{rad}^3$ the relative weights of the second-, third-, and fourth-order terms to $\Phi(\omega)$ are about 9.6, 0.56, and 0.76 rad at the HWHM position of the spectrum (calculated via $|D_n \times (\Delta\omega_{\text{HWHM}})^n/n!|$). This estimation demonstrates that the lowest second-order linear chirp clearly dominates the trend of the non-linear spectral phase, while the impact of D_4 is still minor despite its comparatively large numerical value even for a moderately low pulse energy.

In this context, we added in the main text, caption of Fig. 4, the sentence:

It should be noted that the stated numerical values for the different orders of chirp, in units of fs^2 , fs^3 and fs^4 , are displayed on the same scale on the vertical axis, which does not reflect their relative contribution to the total non-linear chirp (see Methods, Order-of-magnitude estimation of non-linear chirp contributions).

One possible explanation for the origin of the change of D_4 at moderately low fluences may be related to systematics in the process of electron-beam acceleration and compression and its lasing properties, having operated the accelerator not at full compression.

We now give this explanation adding the following passage in the main text on pages 12-13

It should be noted, that the retrieved chirp is clearly dominated by the lowest second-order linear contribution related to D_2 , while the higher orders weigh in less by approximately an order of magnitude (see Methods, Order-of-magnitude estimation of the non-linear chirp contributions, for details). Our direct spectral method is thus sensitive to the non-linear chirp of the measured pulses, evidencing a stronger relative non-linear contribution [i.e., deviation from a purely linear $\tau(\omega)$] at moderately low fluence (cf. Fig. 4). One possible explanation for this observation may be related to systematics in the process of electron-beam acceleration and compression and its lasing properties, having operated the accelerator not at full compression.

In the main text on page 13 we deleted:

This observation may point toward a connection between minimal chirp and optimal electron bunch settings for lasing with new opportunities to maximize the performance and understanding of FELs in the future.

Reviewers' Comments:

Reviewer #2:

Remarks to the Author:

I have reviewed the manuscript entitled "Measuring the frequency chirp of extreme-ultraviolet free electron laser pulses by transient absorption spectroscopy," resubmitted by Ding et al. As I said in my original review, this manuscript is well written and with the additional input from myself and the other referees, it is very clear and understandable. In general, I find this measurement quite interesting. I believe that the effects observed in this measurement will be important to consider in any future pump/probe transient absorption experiments carried out at any FEL facility. Any systematic variation of the x-ray wavelength in time will influence spectrally-dispersed time-resolved measurements. There is high interest from many members of the FEL user community to employ this type of detection scheme, so it is important that the community understands these types of systematic variations in measurements. Given the response of the Authors to my previous comments, and the very positive comments from the other reviewers, I believe that the current manuscript is acceptable for publication in Nature Communications

James P. Cryan

Reviewer #3:

Remarks to the Author:

In the new version of the manuscript the issues I originally raised were properly addressed. The main one was in the definition of the spectral phase and the Authors clarified how this is used in the context of a SASE FEL. The authors have also added details about the beam properties used during the measurements. In my opinion the paper is suitable for publication.

Luca Giannessi